# Changes in individuals' glaucoma progression velocity after IOP-lowering therapy: A systematic review

Vihar Naik[1], Simran Ohri[1], Elise Fernandez📧[2], Jean-Claude Mwanza[1], David Fleischman📧[1]*

**1** Department of Ophthalmology, University of North Carolina, Chapel Hill, North Carolina, United States of America, **2** School of Medicine, University of North Carolina, Chapel Hill, North Carolina, United States of America

* david_fleischman@med.unc.edu

## Abstract

Primary open-angle glaucoma (POAG) is a chronic progressive optic neuropathy often associated with increased intraocular pressure (IOP). Monitoring glaucoma progression velocity, for example, the rate of change in global indices such as mean deviation (MD), is a common way to determine whether functional deterioration has occurred. This systematic review aims to assess changes in glaucoma progression velocity in response to IOP-lowering therapy at an individual level. A systematic review was conducted following Preferred Reporting Items for Systematic Reviews and Meta-Analysis (PRISMA) guidelines. A comprehensive search of PubMed, Cochrane CENTRAL, and ClinicalTrials.gov from database inception through November 12, 2023, was conducted for randomized clinical trials involving patients with POAG, normal tension glaucoma, or progressive ocular hypertension who received IOP-lowering therapy with a target IOP reduction ≥20% from baseline. Included trials were required to report visual field progression velocity information for individual patients before and after intervention. One study was found to meet inclusion criteria and, therefore, synthesis of data and meta-analysis were unable to be performed. The study reports on 139 eyes of 109 patients from the Ocular Hypertension Treatment Study who reached a POAG endpoint. In these patients, the post-treatment rate of MD change ($-0.27 \pm 0.7$ dB/year) was significantly slower than the pre-treatment rate ($-0.51 \pm 0.8$), $P < 0.01$. In addition, the rate of MD change significantly correlated with mean IOP reduction ($p < 0.001$). The singular study demonstrated that IOP-lowering therapy did variably slow glaucoma progression rate in that patient cohort. There is a need for more studies that focus on individual patients' responses to glaucoma treatment. Furthermore, this information should be used to classify the magnitude of patients' responsiveness to IOP reduction. Future studies should report pre- and post-intervention progression velocities.

**Data availability statement:** All relevant data are within the paper and its Supporting Information files.

**Funding:** The author(s) received no specific funding for this work.

**Competing interests:** The authors have declared that no competing interests exist.

## Introduction

Primary open-angle glaucoma (POAG) is a progressive optic neuropathy, frequently linked to elevated intraocular pressure (IOP), which leads to distinctive damage to the optic nerve head and subsequent functional impairment [1]. The pathophysiological mechanisms of POAG remain enigmatic, and clinicians have yet to establish a definitive diagnostic benchmark [2]. Therapeutic strategies aim to halt or markedly decelerate disease progression by reducing IOP. Notably, for a condition to be definitively classified as glaucoma, it typically should show a deceleration in progression following the commencement of IOP-lowering treatment. Although such criteria may not be feasible in routine clinical practice, it is critical for research purposes to accurately differentiate between glaucomatous and non-glaucomatous individuals to maintain the integrity of study outcomes and distinguish the extent of responsiveness to IOP reduction. Regrettably, many studies that encompass glaucoma patients have not rigorously applied such discriminating standards, potentially muddling the research findings with data from patients with varying responses to treatment or even from those without glaucoma.

Serial standard automated perimetry (SAP) testing throughout the patient's lifespan is the most used clinical method for monitoring and detecting glaucoma-related functional deterioration. One common way of determining whether visual field (VF) progression has occurred is to calculate the rate of change (velocity) in global indices (i.e., mean deviation or MD) [3–7]. However, glaucoma progression velocity has been shown to vary highly between individuals, and thus optimal treatment should be individualized and tailored to disease stage and progression rate [8,9]. Even more concerning would be the case in which patients did not respond to IOP reduction whatsoever, perhaps precluding their diagnosis of glaucoma altogether. Without discerning this level of detail, it is likely studies have included patients who have a wide spectrum of response to IOP reduction, likely flustering some of our most highly regarded randomized clinical trials.

Exploring the behavior of glaucoma progression at an individual level, especially in response to initial treatment, may better elucidate underlying mechanisms and potential therapeutic strategies by creating pure or better-differentiated datasets. The term *individual level* signifies that patients meeting entry criteria are analyzed as individuals as opposed to within groups. The present systematic review seeks to better characterize glaucoma progression by evaluating changes in progression velocity at an individual level in response to IOP-lowering therapy.

## Methods

This systematic review was conducted in accordance with the Preferred Reporting Items for Systematic Reviews and Meta-Analysis (PRISMA) statement [10]. The protocol was registered with the International Prospective Register of Systematic Reviews (PROSPERO) (Registration number: CRD42023456208).

### Eligibility criteria

The following criteria were applied to determine eligibility for study inclusion in our review. We included randomized clinical trials of adult (≥18 years of age) patients

diagnosed with POAG, normal tension glaucoma, or ocular hypertension deemed to be progressing. Clinical trials were included regardless of IOP-lowering treatment modality (e.g., medical treatment, laser trabeculoplasty, filtration surgery) and if target IOP reduction was ≥ 20% from baseline.

Included trials had to provide sufficient information regarding progression velocity for individual subjects before and after the intervention, specifically pertaining to VF progression (e.g., rate of MD change). Trials with non-IOP lowering interventions were excluded. Patients with secondary glaucoma, and acute or chronic angle closure were excluded. Studies without information on VF progression or with data from which this could not be reliably extrapolated were also excluded.

## Information sources and search strategy

We conducted a systematic literature search of PubMed, Cochrane CENTRAL, and ClinicalTrials.gov databases from database inception until November 12, 2023. Reference lists from relevant articles were also examined as a supplementary source. We used the following search strategy filtered for clinical trials and randomized clinical trials: (((("glaucoma"[MeSH Terms]) OR (ocular hypertension[MeSH Terms]) AND ((disease progression[MeSH Terms]) OR (progression velocity[All Fields]) OR (progression[All Fields]))) AND (((("treatment"[All Fields]) OR ("therapy"[All Fields])) OR (intraocular pressure[MeSH Terms])).

## Study selection

Study selection was performed with the assistance of Covidence systematic review software (Veritas Health Innovation, Melbourne, Australia). Duplicate records were automatically removed by the software. All remaining original articles and abstracts were screened independently by two reviewers (VN and SO). Any discrepancies were resolved by consensus. Articles satisfying initial screening underwent independent full-text review by VN and SO. Discrepancies were again resolved by consensus, and if required, consultation of senior researchers (DF and JCM).

## Data extraction

The following data were abstracted from each study and tabulated into a spreadsheet: number of patients, study design, inclusion criteria, glaucoma diagnostic criteria, study length, treatment modality, change in progression velocity, and percentage IOP reduction. Two reviewers (SO and EF) independently extracted the data which were subsequently verified by an additional reviewer (VN).

## Risk of bias assessment

For each included study, risk of bias was determined using the revised the Cochrane 'Risk of bias' tool for randomized trials (RoB 2) [12]. RoB 2 addresses five specific domains: 1) bias arising from the randomization process; 2) bias due to deviations from intended interventions; 3) bias due to missing outcome data; 4) bias in measurement of the outcome; and 5) bias in selection of the reported result. Two reviewers (VN and SO) independently performed this assessment and disagreements were resolved by consensus.

## Statistical analysis

We planned to determine effect size for continuous variables using mean differences and associated standard errors calculated from values reported by studies. Synthesis and meta-analysis could not be undertaken because only a single study was ultimately included. P-values less than 0.05 were considered statistically significant.

## Deviations from original protocol

We have included all protocol deviations here to ensure transparency. While our initial protocol specified that Embase would be included in the search strategy, we ultimately did not incorporate it into the final review. A preliminary search of

Embase yielded a high volume of articles that did not align with our inclusion criteria, as many were outside the scope of our research question. Given the limited relevance of the retrieved studies, we determined that Embase was not a suitable database for this review.

Regarding the IOP reduction cutoff, although a benchmark of 30% reduction is commonly used, and we had accordingly set this as our threshold in our original study protocol, a 20% reduction has been shown to be clinically significant in the Ocular Hypertension Treatment Study (OHTS) [11]. Additionally, it is postulated that effects of IOP reduction are likely gradated and not predicated on reduction to an absolute value [8]. Thus, we believed the 20% cutoff to be more appropriate and thereby revised the cutoff used for our analysis. We acknowledge that it would likely increase the breadth of our review.

## Results

### Risk of bias assessment

The assessment of Risk of Bias is presented in Table 1.

A detailed breakdown of signaling questions and domain-specific judgements is provided in Table 2.

### Study selection

Our search strategy initially identified 641 studies for review. Upon import to Covidence, 271 duplicates were identified and removed. The remaining 370 studies were screened, and 3 studies were retrieved for full-text review. Ultimately, only one study met our inclusion criteria and was included for review. The study selection process is detailed in Fig 1. During initial title and abstract screening, inter-rater reliability was measured as kappa of 0.05. However, since there is substantial imbalance in the number of included vs. excluded studies, we suspect this metric is artificially poor. A prevalence adjusted kappa was calculated to be 0.87. For the 3 papers which passed initial screening agreement was 100% with kappa of 1.0 representing perfect agreement. The study selection process is detailed in Fig 1, and data extracted from the included study are summarized in Table 3.

### Study characteristics

The included study followed 780 eyes of 432 patients enrolled in the OHTS [13]. Of these, 139 eyes (17.8%) of 109 participants (25.2%) reached a POAG endpoint. In these study participants, rate of MD change differed significantly before vs.

Table 1. Risk of bias assessment summary based on RoB 2 tool across five domains for the included study.

**Risk of Bias Assessment**

| Domain | Judgement | Reasoning |
|---|---|---|
| 1. Bias Arising from the Randomization Process | Low risk | Randomization was stratified and centrally managed. Allocation concealment was maintained. |
| 2. Bias Due to Deviations from Intended Interventions | Some concerns | Lack of blinding for participants and personnel, though adherence was monitored. |
| 3. Bias Due to Missing Outcome Data | Some concerns | Measures to reduce missing data were included, but details about handling missing data were limited. |
| 4. Bias in Measurement of the Outcome | Some concerns | Standardized outcome assessments were used, but unable to confirm assessor blinding. |
| 5. Bias in Selection of the Reported Result | Low risk | Pre-specified outcomes were clearly defined, reducing the risk of selective reporting. |
| **Overall Judgment** | **Low risk** | **Lack of blinding and limited information on missing data handling introduce some potential bias. However, this bias is unlikely to directionally impact outcomes.** |

Prepared using the revised Cochrane 'Risk of bias' tool for randomized trials (RoB 2) [12].

**Table 2. Domain-specific signaling questions and judgments per the RoB 2 tool for the included study.**

**Risk of Bias Assessment**

| Domain | Signaling Question | | Domain Judgement | Comments |
|---|---|---|---|---|
| **1. Bias Arising from the Randomization Process** | Was the allocation sequence random? | Yes | | |
| | Was the allocation sequence concealed until participants were enrolled and assigned to interventions? | Yes | | |
| | Did baseline differences between intervention groups suggest a problem with the randomization process? | No | | |
| | | | **Low risk** | **Randomization was stratified and centrally managed. Allocation concealment was maintained.** |
| **2. Bias Due to Deviations from Intended Interventions** | Were participants aware of their assigned intervention during the trial? | Probably Yes | | |
| | Were carers and people delivering the interventions aware of participants' assigned intervention during the trial? | Probably Yes | | |
| | Were there deviations from the intended intervention that arose because of the trial context? | Probably No | | |
| | | | **Some concerns** | **Lack of blinding for participants and personnel, though adherence was monitored.** |
| 3. Bias Due to Missing Outcome Data | Were data for this outcome available for all, or nearly all, participants randomized? | No information | | |
| | Is there evidence that the result was not biased by missing outcome data? | No | | |
| | Could missingness in the outcome depend on its true value? | Probably No | | |
| | | | **Some concerns** | **Measures to reduce missing data were included, but details about handling missing data were limited.** |
| 4. Bias in Measurement of the Outcome | Was the method of measuring the outcome inappropriate? | No | | |
| | Could measurement or ascertainment of the outcome have differed between intervention groups? | No | | |
| | Were outcome assessors aware of the intervention received by study participants? | Probably Yes | | |
| | Could assessment of the outcome have been influenced by knowledge of intervention received? | No | | |
| | | | **Some concerns** | **Standardized outcome assessments were used, but unable to confirm assessor blinding.** |
| 5. Bias in Selection of the Reported Result | Were the data that produced this result analysed in accordance with a pre-specified analysis plan that was finalized before unblinded outcome data were available for analysis? | Yes | | |
| | ... multiple eligible outcome measurements (e.g., scales, definitions, time points) within the outcome domain? | No | | |
| | ... multiple eligible analyses of the data? | No | | |
| | | | **Low risk** | **Pre-specified outcomes were clearly defined, reducing the risk of selective reporting.** |
| **Overall Judgment** | | | **Low risk** | **Lack of blinding and limited information on missing data handling introduce some potential bias. However, this bias is unlikely to directionally impact outcomes.** |

Prepared using the revised Cochrane 'Risk of bias' tool for randomized trials (RoB 2) [12].

**Table 3. Summary of data extracted from the included study.**

**Data Extraction Table**

| Study | Extraction | Meets Inclusion Criteria? | Extracted Data | Source |
|---|---|---|---|---|
| De Moraes CG, Demirel S, Gardiner SK, et al. Effect of Treatment on the Rate of Visual Field Change in the Ocular Hypertension Treatment Study Observation Group. Investigative Ophthalmology & Visual Science. 2012;53(4):1704–1709. https://doi.org/10.1167/iovs.11-8186 | Performed by SO and EF on 12/12/2023 | Yes | sample size, study inclusion criteria, glaucoma diagnostic criteria, study length, treatment modality, change in progression velocity (mean deviation pre- and post-intervention), and percentage IOP reduction | Data obtained from the referenced study article as well as from the original published study design [14]. |

after treatment (−0.51±0.8 vs. −0.27±0.7 dB/year, respectively, P<0.01). Additionally, this subgroup demonstrated high individual variability in rate of MD change. A statistically significant correlation was found between rate of MD change and IOP reduction using both the mean IOP and the maximum IOP (all P<0.001). The design and baseline characteristics of the OHTS have previously been described in detail [14]. The characteristics of this study are presented in Fig 2.

## Discussion

This review presents a novel analysis of progression velocity changes following the initiation of IOP-lowering therapy, focused on individual patient data. As far as we are aware, this is the first examination of this topic. Only one study met criteria for inclusion, the Ocular Hypertension Treatment Study (OHTS). In the OHTS, participants with statistically significant elevated IOP yet normal baseline VFs were allocated to either an observation cohort or a treatment pathway [15]. Our review incorporated data from 109 individuals who, although initially assigned to observation, subsequently commenced medical treatment and met the POAG endpoint criteria for inclusion. These individuals had demonstrated two VFs within normal limits before starting therapy, despite the moderately lenient reliability parameters, with false positives, false negatives, and fixation losses being under 33%. The MD rates were calculated by analyzing the six VFs conducted before therapy and the first six VFs taken nine months post-therapy initiation. The results indicated a substantial improvement in the MD rate after the introduction of IOP-lowering therapy, which, in practical terms, suggests a preservation of vision over time. While the specific extent of IOP reduction was not directly measured, the OHTS aimed for a target of 20% reduction [14]. Further, regression analyses revealed a significant association between changes in MD rate and both the average and maximum IOP levels during treatment. This evidence underscores the efficacy of IOP-lowering treatments in decelerating glaucoma progression at an individual patient level.

An intriguing aspect of these findings is the considerable number of patients—nearly half of the cohort—whose rates of progression actually worsened post-treatment initiation. This phenomenon could be indicative of various clinical situations. Some patients may require greater IOP reduction, while others may experience skewed IOP readings due to corneal properties or other biomechanical factors. Additionally, there might be concurrent ocular conditions that progress independently of IOP levels. These variations underscore the complexity of glaucoma as a disease and the challenges in gauging the effectiveness of IOP-lowering treatments. They raise critical questions about the diagnosis and classification of glaucoma. Specifically, it prompts us to consider how many patients diagnosed with glaucoma are truly affected by the condition, and to what extent they benefit from IOP reduction. Furthermore, these observations compel us to reflect on the implications for past studies that have included patients with similar variability in treatment response.

Considering the numerous glaucoma studies reporting on progression of disease, we were surprised that only one met our inclusion criteria. Yokoyama et al. conducted a study which randomized patients with POAG who were already undergoing pressure lowering therapy to be treated with brimonidine or timolol and reported baseline and post-treatment MD slopes [16]. However, patients had already undergone previous aggressive treatment for glaucoma and exhibited

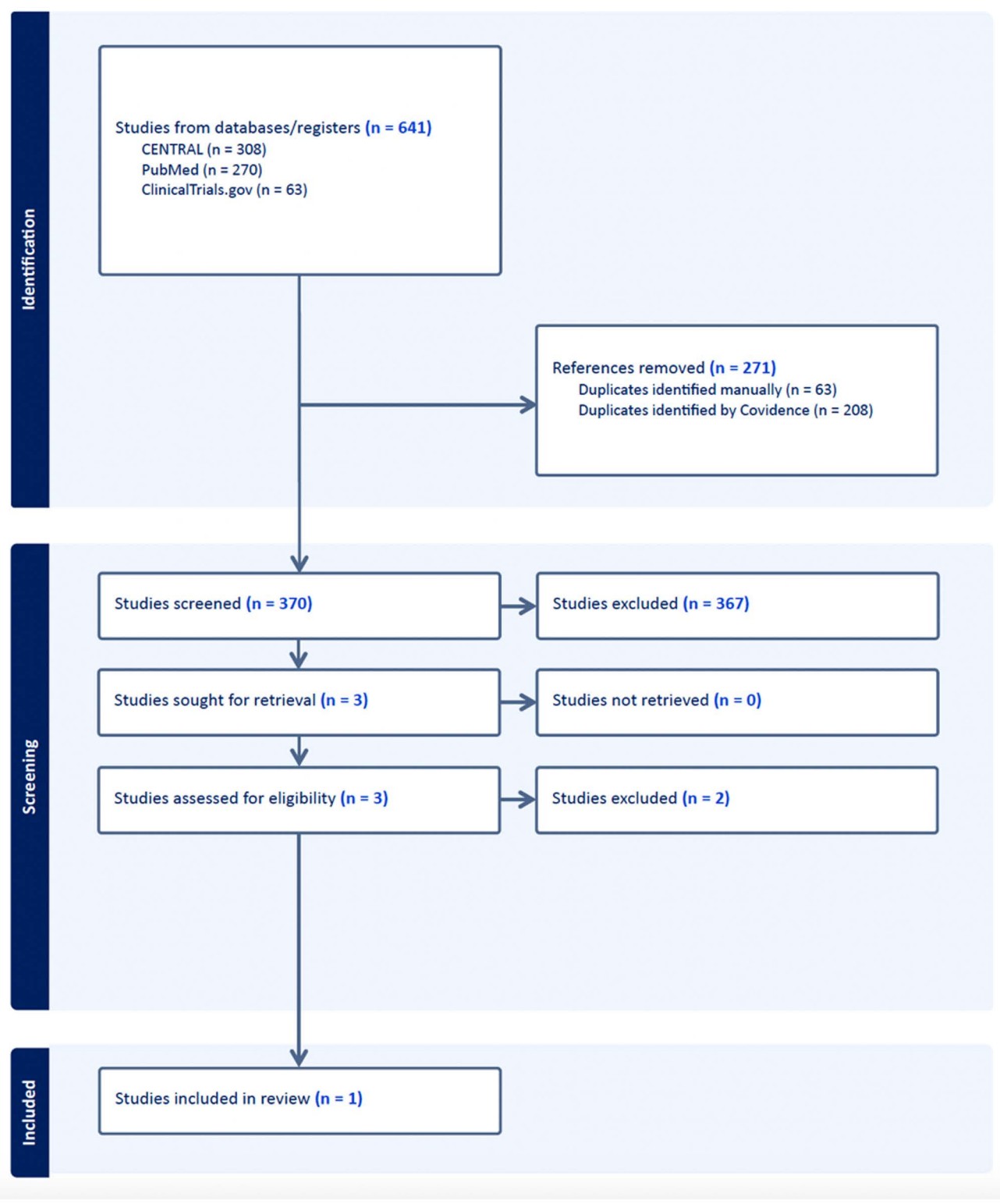

**Fig 1. Study selection flow diagram.**

## Study Characteristics

| Study | Purpose and Design | Disease and Criteria | Progression Velocity Change |
|---|---|---|---|
| De Moraes et al. (13)<br><br>Effect of treatment on the rate of visual field change in the ocular hypertension treatment study observation group | **Purpose:** To compare the velocity of VF progression before and after the initiation of topical hypotensive treatment in subjects from the OHTS.<br><br>**Subjects:** 139 eyes from 109 participants in the OHTS who reached a POAG endpoint<br><br>**Design:** For each subject, the velocities of VF progression were calculated and compared using 6 VFs from before and after initiation of treatment. Velocity was calculated as MDR (dB/year). | **Included:** Participants of the OHTS originally randomized to observation who had a change in management from observation to treatment during the study. Subjects were aged 40-80 years, with no evidence of glaucomatous damage, and with an IOP between 24 and 32 mmHg in one eye and between 21 and 32 mmHg in the other eye.<br><br>**Excluded:** Those that reached non-POAG endpoints and those with unreliable VF tests. | **Before:** $-0.51 \pm 0.8$ dB/year<br><br>**After:** $-0.27 \pm 0.7$ dB/year<br><br>$p < 0.01$ |

VF, visual field
MDR, MD rate of change

**Fig 2. Study characteristics.**

progressive VF defects at baseline. Patients also did not demonstrate >20% IOP reduction and thus this study was excluded from our review.

Multiple other studies reported MD values between groups (e.g., treatment vs control) but not changes in progression velocity after treatment within a group [3,17–20]. These studies are useful in comparatively evaluating therapeutic choices within a certain population but do not help characterize the behavior of glaucoma progression in these populations. Other glaucoma progression studies did provide progression rates during study periods, but did not report baseline progression velocities [6,21,22]. Understandably so, determining baseline progression in a population of untreated patients is difficult and it may even take years to properly establish an accurate progression velocity [7]. The Collaborative Normal Tension Glaucoma Study (CNTGS, n = 230) provided an event-based analysis of progression using survival curves to

operationalize progression [23] in contrast to the trend-based analysis we seek to describe. The United Kingdom Glaucoma Treatment Study (UKGTS) similarly reported time to progression, the proportion of patients who progressed by VF, and the mean IOP reduction [24,25].

These results beg the question that, if glaucoma is a complex and sometimes an obscure entity, to what extent do we need to rely on progression data to properly establish a diagnosis? 68% of participants enrolled in the Early Manifest Glaucoma Trial (EMGT, n = 255) did not demonstrate progression even after a 6-year follow-up [26], a finding echoed by the CNTGS where approximately 55% of participants did not progress after a 6-year follow-up [9]. Ohnell and colleagues additionally later re-assessed VFs and optic disc photos from the EMGT and determined that a small percentage of participants (n = 16, 6.3%) did not meet diagnostic criteria for glaucoma [27]. It is likely that this a is circumstance like this occurs more frequently and affects all glaucoma trials but remains unreported. One must certainly consider that from a clinical perspective, it is difficult to take a patient under high suspicion for glaucoma and wait to document progression prior to initiating treatment, as any interim vision loss would be irreversible. Similarly, in large clinical trials, it may be difficult to find a "clean" dataset from which to learn how glaucoma progresses naturally, as it would be unethical to follow untreated newly-diagnosed patients in order to establish a baseline progression velocity.

Glaucoma varies in severity between individuals, with differences in progression leading to so-called "fast progressors" and "slow progressors" and require stratified diagnoses to best guide treatment [9,21,28]. Further, response to treatment could be considered another important outcome from future studies. A treated patient who exhibits a significant reduction in progression velocity following IOP reduction, regardless of that velocity, could be described as maximally-responsive glaucoma. Patients demonstrating a nominal change in progression velocity could be described as poorly-responsive glaucoma. Patients not demonstrating any change in progression velocity after IOP reduction should not be considered glaucoma under most circumstances. Patients with minimal glaucomatous disease or no response to IOP reduction should be classified as having predominantly glaucoma-like optic neuropathies and investigated as such. Additionally, glaucoma should be ruled out in favor of a neurologic lesion (i.e., optic nerve compression) in the presence of a rapid visual field deterioration and visual acuity despite a good IOP control. It is also possible that glaucoma-like optic neuropathies include many forms of optic nerve damage that are not IOP-responsive but have not yet been characterized. Data from glaucoma investigations utilizing pre-post-treatment progression velocity changes may help identify and elucidate characteristics of these patients.

Limitations of our review include the inclusion of a single study and thus inability to perform a synthesis and subsequent analysis. While many studies do provide MD data for interventions and even MD rates, only one study provided clear cut pre- and post-intervention progression velocities. It is likely that the information we seek is already present in the raw data of many of these studies. Future analyses and trials should seek to better characterize progression velocity changes as they relate to IOP-lowering therapy.

Glaucoma is a complex, heterogenous disease that has been largely treated as a singular entity for over a century. Our goal is to highlight that many landmark glaucoma studies rely on cohort-level data to draw conclusions, which may not always reflect individual responses to treatment. This approach raises the possibility that some studies might have included non-glaucomatous patients, potentially diluting the results. While the current study does not definitively confirm this, it underscores the need for further investigation. Additionally, we propose that certain patients may present with both glaucoma and a glaucoma-like optic neuropathy, warranting more nuanced analysis. This review encourages the use of pre- and post-treatment progression velocities, when practical, in future glaucoma treatment investigations.

## Supporting information

**S1 File. List of studies screened and assessed for eligibility. This document includes all studies reviewed in full text during the systematic review process, along with reasons for inclusion or exclusion.**
(PDF)

**S2 File. PRISMA 2020 checklist. Completed checklist outlining the review's adherence to PRISMA reporting standards for systematic reviews.**
(PDF)

## Author contributions

**Conceptualization:** Jean-Claude Mwanza, David Fleischman.

**Data curation:** Vihar Naik, Simran Ohri, Elise Fernandez.

**Formal analysis:** Vihar Naik, Simran Ohri, David Fleischman.

**Investigation:** Simran Ohri, Elise Fernandez, David Fleischman.

**Methodology:** Vihar Naik, David Fleischman.

**Project administration:** David Fleischman.

**Supervision:** Jean-Claude Mwanza, David Fleischman.

**Visualization:** Vihar Naik.

**Writing – original draft:** Vihar Naik.

**Writing – review & editing:** Jean-Claude Mwanza, David Fleischman.

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
