## [Decision Letter · Decision Letter 0]

29 Nov 2024

Dear Dr. Fleischman,

We look forward to receiving your revised manuscript.

Kind regards,

Petros Cyrus Kayange

Academic Editor

PLOS ONE

Journal Requirements:

2. Please identify your study as "systematic review" in the title of your manuscript

3. We note that your Data Availability Statement is currently as follows: “All relevant data are within the manuscript and its Supporting Information files.”

Please confirm at this time whether or not your submission contains all raw data required to replicate the results of your study. Authors must share the “minimal data set” for their submission. PLOS defines the minimal data set to consist of the data required to replicate all study findings reported in the article, as well as related metadata and methods (https://journals.plos.org/plosone/s/data-availability#loc-minimal-data-set-definition ).

If your submission does not contain these data, please either upload them as Supporting Information files or deposit them to a stable, public repository and provide us with the relevant URLs, DOIs, or accession numbers. For a list of recommended repositories, please see https://journals.plos.org/plosone/s/recommended-repositories .

Reviewers' comments:

Reviewer's Responses to Questions

**Comments to the Author**

1. Is the manuscript technically sound, and do the data support the conclusions?

Reviewer #1: Yes

Reviewer #2: Partly

Reviewer #3: Yes

2. Has the statistical analysis been performed appropriately and rigorously?

Reviewer #1: Yes

Reviewer #2: Yes

Reviewer #3: N/A

3. Have the authors made all data underlying the findings in their manuscript fully available?

Reviewer #1: Yes

Reviewer #2: Yes

Reviewer #3: Yes

4. Is the manuscript presented in an intelligible fashion and written in standard English?

Reviewer #1: Yes

Reviewer #2: Yes

Reviewer #3: Yes

Reviewer #1: This is a well written manuscript raising the issue in the field of glaucoma diagnosis and monitoring. You have covered an important topic when it comes to the glaucoma s a complex disease entity. The statistical analysis were sound. The PRISMA check list was utilised along with Cochrane risk of bias recommendation for the study. Overall, it is a timely and important topic to cover for the individualised treatment approaches for glaucoma patients.

Reviewer #2: 1. The protocol mentions that Embase would be searched, yet the review does not list Embase among the databases used. Could you clarify why Embase was not included in the search strategy? To provide transparency, please consider adding a paragraph titled “Differences Between Protocol and Review” that outlines all deviations from the initial protocol, including the decision to omit Embase. A similar comment applies to the protocol’s statement that the review would use a post-intervention intraocular pressure (IOP) reduction threshold of greater than 30%, whereas the review itself indicates a threshold of greater than 20%. Please clearly justify this change within the proposed paragraph.

2. Please provide the complete search strategy as an appendix to allow readers to replicate or verify the search methodology.

3. Could you also report on the inter-rater reliability for study inclusion decisions? This would strengthen the review’s methodological rigor. Or explain the reason for not including it in the manuscript.

4. In the discussion section, when referencing the Early Manifest Glaucoma Trial (EMGT) and the Collaborative Normal Tension Glaucoma Study (CNTGS), it would be helpful to include cohort numbers directly. This will save readers the need to look up these figures in the original articles. Furthermore, in the EMGT, please specify the exact percentage of participants who did not meet the diagnostic criteria for glaucoma, as this data provides important context to your findings.

5. In the future if you will consider an update on this review, it may be valuable to consider contacting study authors for raw data. This could be an effective approach if you suspect such data is available but was not accessible in the initial review.

6. Regarding your statement in the manuscript’s conclusion—"Our hope is to raise awareness that the data are not conclusive regarding an individual's response to glaucoma treatment, that many landmark studies may have included non-glaucomatous patients, potentially diluting results, and that some patients may have glaucoma and a glaucoma-like optic neuropathy"—if this indeed reflects the review’s conclusion, I would note that the review does not present sufficient evidence to substantiate these claims. I recommend revisiting and reworking the conclusion to better align it with the review’s findings or to support this assertion with additional evidence.

7. Additionally, while the risk of bias methodology is well-described on page 6, the actual results of this assessment were not provided. Including these results in the submission would provide readers with a clearer understanding of the potential limitations and strengths.

8. Finally, I consider it important that articles from the journal to which the manuscript is submitted are cited, as this adds to its relevance within the journal's academic discourse.

Reviewer #3: Many thanks to the Editor for an opportunity to review this paper. The question raised by the authors is important. However, for me remains unclear what the difference between “individual level” and general relationships known for IOP reduction and glaucoma progression. In their systematic review authors were able to identify a single study (OHTS) met inclusion criteria. In such case, which new information provides this paper compared to the above mentioned study? The fact that such studies are underreported is important but definitely cannot be a main conclusion of the review.

**Do you want your identity to be public for this peer review?** For information about this choice, including consent withdrawal, please see our Privacy Policy

Reviewer #1: **Yes: ** Pouya Alaghband

Reviewer #2: No

Reviewer #3: No

---

## [Author Response · Author response to Decision Letter 1]

2 Apr 2025

Response to Reviewers’ Comments

Manuscript Title: Changes in individuals’ glaucoma progression velocity after IOP-lowering therapy

Dear Editor,

We sincerely appreciate the time and effort of the reviewers and the editorial team in assessing our manuscript. We have carefully considered each comment and have revised the manuscript accordingly. Below, we provide a point-by-point response to all comments, highlighting the changes made. We hope that these revisions enhance the clarity and rigor of our work.

Sincerely,

David Fleischman and Study Team 

Response to Reviewer #1

Comment 1:

Reviewer: This is a well written manuscript raising the issue in the field of glaucoma diagnosis and monitoring. You have covered an important topic when it comes to the glaucoma s a complex disease entity. The statistical analysis were sound. The PRISMA check list was utilised along with Cochrane risk of bias recommendation for the study. Overall, it is a timely and important topic to cover for the individualised treatment approaches for glaucoma patients.

Response:

We thank the reviewer for their encouragement and appreciate their recognition of the importance of this topic.

Response to Reviewer #2

Comment 1:

Reviewer: The protocol mentions that Embase would be searched, yet the review does not list Embase among the databases used. Could you clarify why Embase was not included in the search strategy? To provide transparency, please consider adding a paragraph titled “Differences Between Protocol and Review” that outlines all deviations from the initial protocol, including the decision to omit Embase. A similar comment applies to the protocol’s statement that the review would use a post-intervention intraocular pressure (IOP) reduction threshold of greater than 30%, whereas the review itself indicates a threshold of greater than 20%. Please clearly justify this change within the proposed paragraph.

Response:

We sincerely thank the reviewer for their careful attention to this important detail. We acknowledge that Embase was initially included in the protocol but was ultimately not used in the review. Our preliminary search in Embase yielded articles that were largely not aligned with our inclusion criteria, leading us to determine that it was not a suitable database for this review. To ensure transparency, we have added a section titled “Deviations from original protocol" in the Results section, where we justify this deviation, as well as the IOP reduction threshold.

Comment 2:

Reviewer: Please provide the complete search strategy as an appendix to allow readers to replicate or verify the search methodology.

Response:

We have provided the search strategy below as outlined in the Methods section, since it is an integral part of the study’s methodology.

Sources: PubMed, Cochrane CENTRAL, and ClinicalTrials.gov

Time Frame: inception until November 12, 2023.

Terms: (((“glaucoma"[MeSH Terms]) OR (ocular hypertension[MeSH Terms]) AND ((disease progression[MeSH Terms]) OR (progression velocity[All Fields]) OR (progression[All Fields]))) AND ((("treatment"[All Fields]) OR ("therapy"[All Fields])) OR (intraocular pressure[MeSH Terms])).

Comment 3:

Reviewer: Could you also report on the inter-rater reliability for study inclusion decisions? This would strengthen the review’s methodological rigor. Or explain the reason for not including it in the manuscript.

Response:

We appreciate the Reviewer’s comment. We have added the below paragraph to the manuscript in the Results section.

“During initial title and abstract screening, inter-rater reliability was measured as kappa of 0.05. However, since there is substantial imbalance in the number of included vs excluded studies, we suspect this metric is artificially poor. A prevalence adjusted kappa was calculated to be 0.87. For the 3 papers which passed initial screening agreement was 100% with kappa of 1.0 representing perfect agreement.”

Comment 4:

Reviewer: In the discussion section, when referencing the Early Manifest Glaucoma Trial (EMGT) and the Collaborative Normal Tension Glaucoma Study (CNTGS), it would be helpful to include cohort numbers directly. This will save readers the need to look up these figures in the original articles. Furthermore, in the EMGT, please specify the exact percentage of participants who did not meet the diagnostic criteria for glaucoma, as this data provides important context to your findings.

Response:

Thank you kindly for these suggestions. We have made the requested edits in the manuscript.

Comment 5:

Reviewer: In the future if you will consider an update on this review, it may be valuable to consider contacting study authors for raw data. This could be an effective approach if you suspect such data is available but was not accessible in the initial review.

Response:

We certainly agree and hope the study teams would be open to these requests.

Comment 6:

Reviewer: Regarding your statement in the manuscript’s conclusion—"Our hope is to raise awareness that the data are not conclusive regarding an individual's response to glaucoma treatment, that many landmark studies may have included non-glaucomatous patients, potentially diluting results, and that some patients may have glaucoma and a glaucoma-like optic neuropathy"—if this indeed reflects the review’s conclusion, I would note that the review does not present sufficient evidence to substantiate these claims. I recommend revisiting and reworking the conclusion to better align it with the review’s findings or to support this assertion with additional evidence.

Response:

We sincerely thank the reviewer for this suggestion. As such, we have amended our Conclusion. Please note our revised conclusion:

“Glaucoma is a complex, heterogenous disease that has been largely treated as a singular entity for over a century. Our goal is to highlight that many landmark glaucoma studies rely on cohort-level data to draw conclusions, which may not always reflect individual responses to treatment. This approach raises the possibility that some studies might have included non-glaucomatous patients, potentially diluting the results. While the current study does not definitively confirm this, it underscores the need for further investigation. Additionally, we propose that certain patients may present with both glaucoma and a glaucoma-like optic neuropathy, warranting more nuanced analysis. This review encourages the use of pre- and post-treatment progression velocities, when practical, in future glaucoma treatment investigations.”

Comment 7:

Reviewer: Additionally, while the risk of bias methodology is well-described on page 6, the actual results of this assessment were not provided. Including these results in the submission would provide readers with a clearer understanding of the potential limitations and strengths.

Response:

Thank you so much for this suggestion. We have provided an additional document which includes our risk of bias summary table and its mention within the Results section.

Comment 8:

Reviewer: Finally, I consider it important that articles from the journal to which the manuscript is submitted are cited, as this adds to its relevance within the journal's academic discourse.

Response:

We thank the reviewer for this comment, which we agree with. We are not aware of any manuscripts from PLoS One that we have failed to include here. We are receptive to this information, however, and will enthusiastically include this if you let us know which ones.

Response to Reviewer #3

Comment 1:

Reviewer: Many thanks to the Editor for an opportunity to review this paper. The question raised by the authors is important. However, for me remains unclear what the difference between “individual level” and general relationships known for IOP reduction and glaucoma progression.

Response:

The term “individual level” signifies that patients meeting entry criteria are analyzed as individuals as opposed to within groups. In other words, each datapoint (patient or eye) is sufficiently granular such that we could assess progression velocity before and after treatment. Patients who are similar and clumped into a group may have differential responses to treatment. Some may not have any response to treatment, in fact, while others may have worsening after treatment. This is another way of interpreting the data from the sole study that provided this level of granularity, although their conclusions focused on the macro trend of improvement after IOP reduction.

We have now included a clarifying statement on the terminology “individual level” within the Introduction to reinforce its intent and meaning. We greatly appreciate the Reviewer’s question as others would likely have had the same.

Comment 2:

Reviewer: In their systematic review authors were able to identify a single study (OHTS) met inclusion criteria. In such case, which new information provides this paper compared to the above mentioned study? The fact that such studies are underreported is important but definitely cannot be a main conclusion of the review.

Response:

Thank you for your thoughtful comment. The purpose of our systematic review was to synthesize studies examining progression velocity before and after treatment at an individual level, with the goal of understanding the relationship between treatment response and glaucoma progression. While we did not anticipate that only a single study (OHTS) would meet these criteria, this finding itself is highly significant.

Resubmission Request:

Please identify your study as "systematic review" in the title of your manuscript.

We have changed the title of our manuscript to reflect this was a systematic review.

We note that your Data Availability Statement is currently as follows:

"All relevant data are within the manuscript and its Supporting Information files."

Please confirm at this time whether or not your submission contains all raw data required to replicate the results of your study. Authors must share the “minimal data set” for their submission. PLOS defines the minimal data set to consist of the data required to replicate all study findings reported in the article, as well as related metadata and methods

This is true.

Please include your tables as part of your main manuscript and remove the individual files. Please note that supplementary tables (should remain/ be uploaded) as separate "Supporting Information" files.

Thank you – we have included the Risk of Bias Assessment into the manuscript itself.

---

## [Editor Report · Decision Letter 1]

2 May 2025

Changes in individuals’ glaucoma progression velocity after IOP-lowering therapy: A Systematic Review

PONE-D-24-43557R1

Dear Dr. Fleischman,

We’re pleased to inform you that your manuscript has been judged scientifically suitable for publication and will be formally accepted for publication once it meets all outstanding technical requirements.

Kind regards,

Petros Cyrus Kayange

Academic Editor

PLOS ONE
---

## [Editor Report · Acceptance letter]

PONE-D-24-43557R1

PLOS ONE

Dear Dr. Fleischman,

I'm pleased to inform you that your manuscript has been deemed suitable for publication in PLOS ONE. Congratulations! Your manuscript is now being handed over to our production team.

Kind regards,

on behalf of

Assoc. Professor Petros Cyrus Kayange

Academic Editor

PLOS ONE